# Preliminary Design and Cross-Sectional Form Study of Closed-Type Concrete-Filled Steel Tube Support for Traffic Tunnel

**Lei Li [1] and Ke Lei [2],***

[1]    Bridge and Tunnel Research Center, Research Institute of Highway Ministry of Transport, Beijing 100088, China; li.lei@rioh.cn

[2]    Key Laboratory of Urban Underground Engineering of Ministry of Education, Beijing Jiaotong University, Beijing 100044, China

*    Correspondence: 18115025@bjtu.edu.cn; Tel.: +86-186-9310-4975

**Abstract:** In view of the structural form and common construction methods of traffic tunnels, the bearing performance of the closed-type CFST support designed for traffic tunnels is studied. The closed-type CFST support, which consist of a CFST girder with external shotcrete, is improved from the CFST support used in mine roadways. The reasonable cross-sectional form of closed-type CFST support is analyzed by the FEM. The closed-type CFST support is mainly composed of CFST arches, a shotcrete layer, sleeves, and blind flanges. The post-buckling analysis of the closed-type CFST circular arch members using circular-shaped, rectangular-shaped, triangular-shaped, and trapezoidal-shaped steel tubes is implemented. The result shows that the closed-type CFST support has better performance than the traditional tunnel support. The study also found that for closed-type CFST support, the triangular-shaped steel tube section has the highest bearing capacity, stiffness, and steel utilization rate, which is the preferred cross-sectional form. The bearing capacity of the circular-shaped steel tube section is acceptable. Moreover, the circular-shaped steel tubes are more convenient to obtain and process, so it is also an optional cross-sectional form. The square-shaped and trapezoidal-shaped steel tube sections have neither performance advantages nor economic efficiency, so these two forms are not recommended.

**Keywords:** underground engineering; traffic tunnel; primary support; concrete-filled steel tube; bearing performance

## 1. Introduction

Concrete-filled steel tube (CFST) refers to a member formed by filling concrete with steel tubes. It is a main structural form of steel–concrete compound structures, which is often used as a compression and bending bearing member. The core idea of CFST is to use the restraint effect of the steel tube to improve the bearing capacity and ductility of concrete, so as to amplify the advantages of steel and concrete, and achieve better working performance.

In the construction of the railway piers in Severn, England in 1879, the first attempt was to fill the concrete with steel pipe columns to prevent corrosion of the steel pipes and share part of the pressure [1]. In 1897, the American engineer John Lally filled concrete with round steel tubes as a load-bearing column (called Lally column) of a house building and obtained a patent. This case is considered to be the first application of CFST as a composite structural member [2]. After 1960s, countries and regions such as the former Soviet Union, Western Europe, North America, and Japan carried out a lot of experimental research and engineering applications, but at the time the problem of on-site pouring technology was not well resolved, which caused people to give up the CFST temporarily,

and turned to the steel structure. In the late 1980s, on the one hand, the problems existing in the on-site concrete pouring were solved due to the development of advanced pumping technology; on the other hand, the restraining effect of the steel tube can effectively reduce the brittleness of high-strength concrete [3]. Thus, people focused on the CFST technology again. The CFST technology developed rapidly. The CFST structure has high bearing capacity, good plasticity and toughness, and is convenient for construction. Now the CFST has been widely used in structures such as arch bridges, subways, high-rise buildings, and industrial plants.

CFST were mostly used for compression members at first, so the research on its compression performance is well developed. Choi Chang Sik et al. [4] experimentally studied the axial compressive properties of steel fiber CFST short columns. The result shows the failure modes of the steel fiber CFST short column include local buckling, global-local mixed buckling, and welding failure. The incorporation of steel fibers significantly improves the overall performance of the CFST short column. Hassanein M.F. et al. [5] studied the structural performance of CFST elliptical columns under pure axial compression and eccentric loads through numerical simulations. The article also proposed a design formula considering the effective strength of confined concrete. Based on a large number of test samples, Mansouri Ali [6] proposed a calculation method for the shear strength of CFST. This method believes that the shear strength of CFST equals the total shear strength of steel tubes and core concrete. The parameters that affect the shear strength of concrete include constraint coefficient, shear-span ratio and axial compression ratio. The shear strength of CFST calculated by this method matched well with the test results. Ouyang Yi et al. [7] applied the analytical lateral axial strain relation established by Dong et al. to the finite element analysis model of a CFST column under axial compression. This method can directly calculate the lateral strain of confined concrete, and it avoids the underestimation of the lateral expansion effect while using the traditional plastic theory. The method can accurately predict the peak load and post-peak performance of the rectangular CFST column.

Some researchers have also analyzed the mechanical properties of CFST as a bending member. Cho Jung Hyun et al. [8] derived the formula for calculating the bending strength of CFST composite beams based on the plastic stress distribution method (PSDM), and verified the accuracy of the formula through experiments. The article then studied the influence of different structural parameters on the bending strength of CFST composite beams. Qu Guanglong [9] conducted a four-point bending test on a set of circular CFST beams partially reinforced by round steel. The test verified that the CFST beam satisfies the flat section assumption while it is in pure bending deformation, and summarized the law of the neutral layer deviation during the loading process. The experiment results of the reinforced beams show that the outer welding round steel has the most obvious effect on improving the bending bearing capacity of the CFST beam. Liu Keming [10] used a six-point uniform loading scheme to test and study 10 CFST arches with different rapture ratios, different concrete strengths, and different volumetric contents of steel fibers. The research reveals that during the loading process, the steel tube and core concrete at the crown are always under compression, and no neutral layer deviation occurred. The in-plane load-bearing capacity of CFST arches is closely related to the rapture ratio. The failure of the CFST arch first appeared at the arch foot, which appears as the expansion bulge of the steel tube. The CFST arch has strong resistance to in-plane load and deformation. Therefore, the CFST arch can meet the requirements of mine roadway support.

In view of the excellent performance of CFST, many researchers began to use CFST for the construction of underground projects. Huang W.P. et al. [11] developed a CFST support used in coal mine roadways with a buried depth of more than 1000 m. The load-bearing performance of CFST support is 3–5 times that of U-shaped steel arches. The CFST support and the anchor cable system effectively control the large deformation of soft rock with high ground stress. Since 2004, the team of Professor Gao Yanfa [12] of China University of Mining and Technology has made many achievements in the research of mining CFST support, and has entered the application stage since 2008. The CFST support has demonstrated the excellent performance in more than 20 mines with different geological conditions [13,14]. The team of Professor Li Shucai [15] of Shandong University also

proposed a high-strength CFST support system suitable for tunnels, coal mines, or other underground chambers with a different cross-sectional shape. The CFST support system has been successfully applied in tunnels with extremely unfavorable conditions, such as high geostress, extremely soft rock, strong mining influence, or through a fault fracture zone [16,17], which proves that the CFST support has a good effect in rock stabilization as well as a better economy effect.

In traffic tunnels of the time, primary support with lattice girder or profile steel as the rigid frame is used commonly. When the high ground stress, weak surrounding rocks, and other special geological conditions are encountered, the traditional primary support are easily twisted, rolled, and partially buckled. The CFST has high strength, high stiffness, and excellent resistance to buckling and torsion. Therefore, using CFST as the rigid frame for primary support could be a feasible way to solve this problem. This paper aims to study the bearing performance of the closed-type CFST support designed for traffic tunnels. The closed-type CFST support, which consist of a CFST girder with external shotcrete, is improved from the CFST support used in mine roadways. Through the FEM analysis of the circular arch members abstracted from the traffic tunnel support, some preliminary conclusions about the optimal cross-sectional form of closed-type CFST support have been made.

However, it is necessary to point out that the conclusions in this article are based on some simplifications and assumptions under ideal conditions. The working state of the actual tunnel support is more complicated. Many issues such as the applicable conditions of closed-type CFST support are worthy of further study.

## 2. Conceptual Design of Closed-Type CFST Support

### 2.1. Basic Component of Closed-Type CFST Support

The closed-type CFST support includes the following main parts (shown in Figure 1):

1. CFST girder: The rigid skeleton and one of the main bearing parts of the supporting structure. It is formed by several sections of arc-shaped steel tubes filled with concrete, connected by casings, flanges, and other connecting parts. When the CFST girder is required to be installed section by section, the end of the girder should be connected with a flange and temporarily closed by a blind flange. After the core concrete is initially solidified, it can be connected to the steel tube of the next section after removing the blind flange.

2. Shotcrete layer: Another main bearing part of the supporting structure. The shotcrete layer can make the support and the surrounding rock fit closely. The shotcrete layer is generally applied in two steps. The first layer of shotcrete has the function of stabilizing and protecting the surface of the surrounding rock in time, and at the same time acting as a force-transmitting coupling material between the CFST girder and the surrounding rock. The thickness of the first layer is generally not less than 4 cm. The second layer of shotcrete is applied after the initial solidification of the core concrete. The shotcrete should be sprayed in layers and sections, densely fill all of the gaps between the steel tube and the first shotcrete layer. The second layer should cover the entire supporting structure.

3. Sleeves: The sleeves are the connecting part of the CFST girder. The length of the sleeve is generally about 1 m, and its wall thickness can be the same as or slightly thinner than the constrained steel tube. The sleeve connection will not block the flow of concrete inside the CFST girder, so it is not suitable for the connection of the construction joint.

4. Flange/blind flange: Flange/blind flange is another kind of connection component. Unlike the sleeve, the flange can be temporarily blocked by adding a blind flange, so that the CFST girder can be filled with the core concrete in sections to match various tunnel construction methods.

5. Longitudinal strut: The role of the longitudinal strut is to improve the integrity of the structure and the ability to resist out-of-plane loads. If the craftsmanship permits, the longitudinal strut can also be a CFST beam with a smaller diameter.

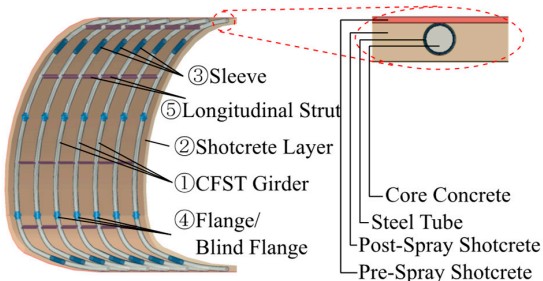

**Figure 1.** Schematic of the closed-type concrete-filled steel tube (CFST) support.

### 2.2. Cross-Sectional Form and Its Design Principles of Closed-Type CFST Support

The cross-sectional form of CFST is mainly determined by the cross-sectional shape of the steel tube. CFST was first used as a compressive member in a building structure, its main purpose is to restrain the core concrete by the ferrule effect of the steel tube, meanwhile to improve the stress state of the core concrete, as to get better bearing capacity and ductility. The circular steel tube has the most significant ferrule effect, so the traditional CFST is mainly circular-shaped. As for the tunnel support, it must have sufficient resistance to bending and compression, as well as good co-working performance with surrounding strata and other supporting members. Regarding the structural form, construction method, and mechanical behavior of the traffic tunnel, closed-type CFST support should fulfill the following requirements:

- It should have reliable working performance under a compression or compression-bending force, and has the ability to resist bidirectional alternating bending. To fulfill this requirement, the section must have a certain tensile strength on both sides of the neutral axis. Therefore, the steel tube should have sufficient cross-sectional area on both sides of the neutral axis.
- At the initial stage of construction, it should be able to withstand the surrounding rock load through the presprayed shotcrete layer, which means the steel tube should have a sufficiently large contact area with the presprayed shotcrete layer.
- The steel tube and the shotcrete should have good cooperative performance and constructability. So, the shape of the steel tube should not cause excessive dead angles when spraying the shotcrete, in order to ensure the compactness of shotcrete.
- Has a certain ability to resist out-of-plane loads and torsional effects.

According to the principles above, the following cross-sectional forms of closed-type CFST support are initially selected for comparative study:

- Circular-shaped: The circular-shaped CFST has the best restraining effect on the core concrete, and it has the same bending resistance in all directions. Its ability to resist torsion and resist local buckling is also excellent. In addition, the circular steel tube and its accessories are easy to obtain, so the application of circular-shaped CFST is the most extensive. However, the circular-shaped CFST also have the disadvantages of relatively large dead space for shotcrete spraying.
- Rectangular-shaped: Like the circular-shaped CFST, the rectangular-shaped CFST also has excellent torsional resistance. The biggest advantage of the rectangular-shaped CFST compared to the circular-shaped is that it has a larger contact area with the presprayed shotcrete, so its early bearing performance will be better, theoretically. However, some studies have pointed out that the restraining effect of the square steel tube is less obvious, and the stress is unevenly distributed along the tube wall, which is prone to stress concentration and local buckling instability. Moreover, the manufacturing difficulty of the rectangular-shaped CFST is also higher than that of circular-shaped CFST.

- Triangular-shaped: The triangular-shaped CFST is similar to the three-legged lattice girder commonly used in tunnel support. Its main advantages are that the steel utilization rate is much higher than other forms of steel tube. Theoretically, it is easier to ensure the compactness of shotcrete. However, the bending resistance in different directions of triangular-shaped CFST is different, which is not ideal for the tunnel support that alternately undergoes bidirectional bending. In addition, there are few studies on the application of triangular-shaped CFST, so its mechanical properties are still in doubt. The difficulty of steel tube processing and concrete pouring of the triangular-shaped CFST is also likely to be higher than other cross-sectional forms.

- Trapezoid-shaped: Trapezoid-shaped is a cross-sectional form between rectangular and triangular, and has the advantages of both. The concept of trapezoidal-shaped CFST is similar to U-shaped and D-shaped CFST, so it can actually be regarded as the same type. Theoretically, the well-designed trapezoidal-shaped CFST may get an excellent balance between performance and economy. However, the trapezoidal-shaped CFST also has many problems. First, the trapezoidal-shaped steel tube is more difficult to manufacture. Second, the design of the trapezoidal-shaped CFST lacks experienced guidance.

The schematic diagram of each cross-sectional form is shown in Figure 2. $D_S$ is the steel tube outer diameter of circular-shaped CFST; $h_S$ and $w_S$ are the steel tube height and width, respectively, of rectangular-shaped, triangular-shaped, and trapezoidal-shaped CFST; $\alpha$ is the interior angle of triangular-shaped and trapezoidal-shaped CFST; $t$ is the steel tube thickness; and $H$ is the total cross-sectional height of the support.

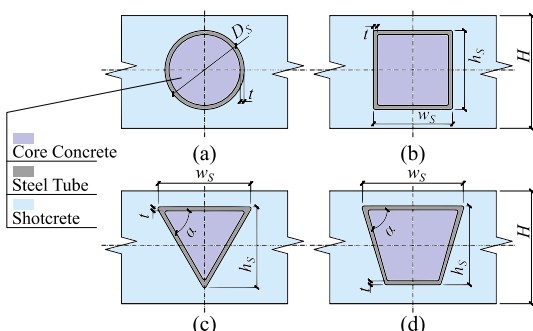

**Figure 2.** Schematic of closed-type CFST support with various cross-sectional forms. (**a**) Circular-shaped; (**b**) rectangular-shaped; (**c**) triangular-shaped; and (**d**) trapezoidal-shaped.

In fact, in addition to the four forms listed above, there may be many other cross-sectional forms for closed-type CFST support. The cross-sectional form has a great influence on the bearing performance, economic performance, and the manufacturing complexity of closed-type CFST support. Therefore, the selection of the cross-sectional form should be given the highest priority during the designing process. In order to make the follow-up research more targeted, it is necessary to analyze and compare the comprehensive performance of closed-type CFST support with various cross-sectional forms. Due to space limitations, this article only studies the four cross-sectional forms listed above.

## 3. Research Plan of Cross-sectional Form Selection

### 3.1. Finite Element Model

#### 3.1.1. Abstraction of Primary Support

Generally speaking, the tunnel support is a cylindrical structure mainly subjected to bending and compression, and the stress state in its ideal state is close to the plane strain condition. In order to simplify the calculation, under the premise of keeping the main mechanical characteristics of the support as much as possible, the abstraction is made with the support of its upper half cross-section.

The circular arch member with a central angle of 120° is regarded as the research object, as shown in Figure 3. *R* is the radius of the member axis, and *B* is the cross-sectional width of the member.

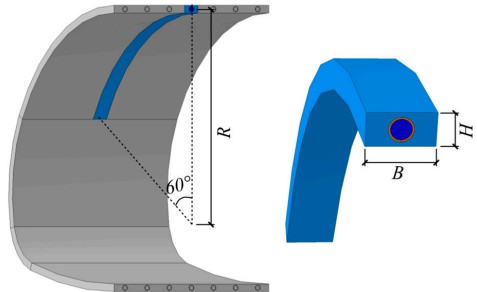

**Figure 3.** Abstraction of primary support.

The thickness of support commonly used in Chinese traffic tunnels is generally 22–30 cm, so the median value of 26 cm is selected as the cross-sectional height *H*. The section width *B*, theoretically, can be any value between the steel tube width $w_S$ and the distance of two adjacent steel tubes. Since this study mainly compared different cross-sectional forms of the steel tube, the abstracted member should contain as little shotcrete as possible to highlight the role of steel tubes. Thus, the cross-sectional width *B* of the abstract member was initially taken as *B* = *H* = 26 cm. The radius of the circular arch *R* was considered as 8.41 m according to the typical half-span of the Chinese three-lane highway tunnel.

### 3.1.2. Model and Mesh Discretization

The finite element modeling was done by the commercial software ABAQUS™. Since special conditions such as unsymmetrical loading were not considered, a 1/4 model was established. The meshing properties are listed as follows:

- The 8-node linear solid elements (C3D8R) were used throughout the model.
- The core concrete, steel tube, and shotcrete were meshed respectively.
- The steel tube (or the profile steel girder) had an approximate grid size of 10 mm on its cross-section and 50 mm along its axis.
- The core concrete had an approximate grid size of 8~20 mm on its cross-section and 40 mm along its axis.
- The shotcrete had an approximate grid size of 15 mm on its cross-section and 80 mm along its axis.

The model geometry and the mesh are shown in Figures 4 and 5.

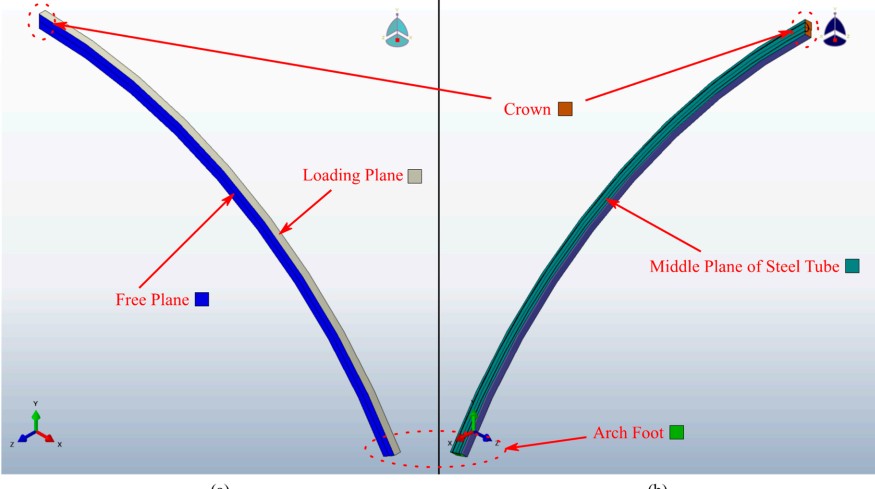

**Figure 4.** Geometry of the finite element model. (**a**) Isometric view and (**b**) reversed isometric view.

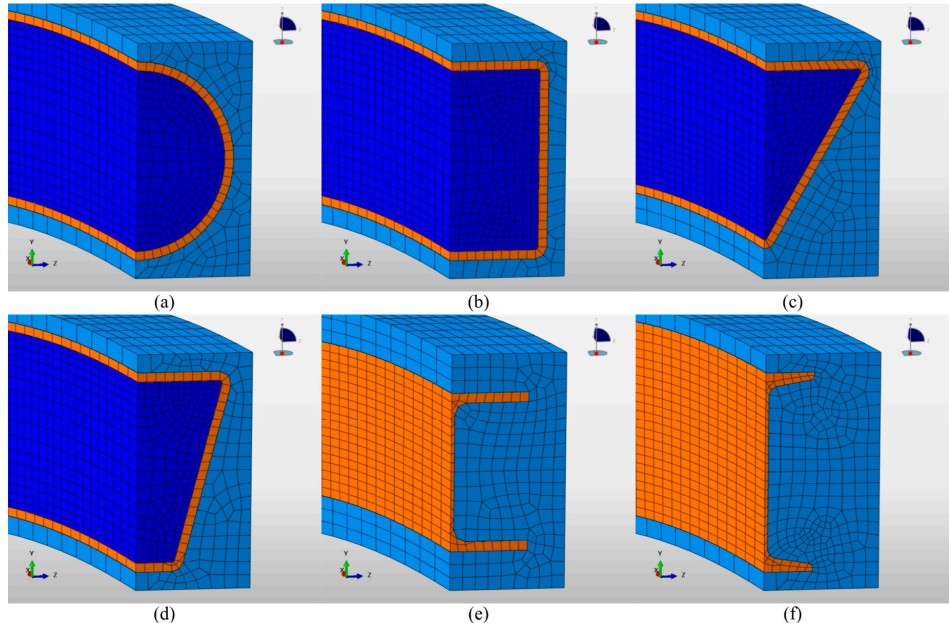

**Figure 5.** Schematic diagram of the finite element mesh. (**a**) Circular-shaped CFST member; (**b**) rectangular-shaped CFST member; (**c**) triangular-shaped CFST member; (**d**) trapezoidal-shaped CFST member; (**e**) H175 profile steel member; and (**f**) I22a profile steel member.

### 3.1.3. Boundary Conditions

The boundary conditions of the model are as follows:

- The top surface of the model (refer to the brown-colored zone in Figure 4) corresponds to the symmetry plane at the crown of the circular arch member. The X-direction symmetry boundary condition for the solid element (constraint X-direction translational freedom) was adopted for the nodes on this surface.
- The bottom surface of the model (refer to the green-colored zone in Figure 4) corresponds to the junction point between the crown and the bench when the tunnel is excavated by the bench method. Before the excavation of the bench, in order to temporarily stabilize the support, foot anchors are usually applied at this point to provide horizontal and vertical restraint. Therefore, the foot of the arch member can be regarded as a fixed point. According to this assumption, rigid constraints (constraint the translational freedom in all spatial directions) were adopted for the nodes on the bottom face of the member.
- The outer surface of the arch (refer to the grey-colored zone in Figure 4) corresponds to the contact surface between the surrounding rock and the support, so it is the surface that the loads acting on. According to the Chinese standard [18], when there are no asymmetry factors such as bias pressure, the distribution of the load near the crown area is very close to the hydrostatic pressure state. Therefore, a uniformly distributed load was adopted for the normal direction of this surface, in order to simulate the load of the tunnel support in the most general case. The value of the load was automatically calculated by the arc length method, which will be mentioned later.
- The boundary surface of the Z-negative direction (refer to the cyan-colored zone in Figure 4) corresponds to the symmetry plane defined by the cross-sectional center line and the member axis. Thus, the Z-direction symmetry boundary condition was adopted for this surface.
- The boundary surface of Z-positive direction (refer to the blue-colored zone in Figure 4) is a free surface.
- When CFST is used as a bending member, studies have proved [17] that the bond-slip has little effect on the overall mechanical response of the member. Thus, tie constraints were established at the interface between the steel tube and the concrete, without considering the bond-slip effect.

### 3.1.4. Solution Scheme

In order to obtain the complete process from elastic deformation to buckling, the calculation adopted the arc length method [19] to analyze the post-buckling behavior of the member. The analysis process is:

1.  Establish the model and impose all boundary conditions except the load.
2.  Apply gravity and perform a general static analysis to obtain the equilibrium state of the member under its own weight.
3.  Apply the load mentioned in Section 3.1.3. Then perform the arc length analysis. The termination condition of the analysis is that the crown settlement reaches 2 m.

### 3.2. Constitutive Model and Material Parameter

#### 3.2.1. Concrete

For the concrete material, an ABAQUS$^{\text{TM}}$ built-in constitutive model, the damaged plasticity model, was adopted. This model was based on the models proposed by Lubliner et al. [20] and Lee and Fenves [21]. The stress–strain relations are governed by scalar damaged elasticity:

$$\sigma_{ij} = (1-d)D^0_{ijkl}\left(\varepsilon_{kl} - \varepsilon^{pl}_{kl}\right) \tag{1}$$

where $D^0_{ijkl}$ is the component of the initial elastic stiffness tensor. $d$ is the scalar stiffness degradation variable. $\sigma_{ij}$ is the component of Cauchy stress tensor. $\varepsilon_{kl}$ and $\varepsilon^{pl}_{kl}$ are the component of total strain tensor and the plastic strain tensor, respectively.

Following the usual notions of continuum damage mechanics, the effective stress tensor is defined as:

$$\overline{\sigma}_{ij} \overset{\text{def}}{=} D^0_{ijkl}\left(\varepsilon_{kl} - \varepsilon^{pl}_{kl}\right) = \sigma_{ij}/(1-d) \tag{2}$$

The scalar stiffness degradation variable, $d$, is defined as

$$d = 1 - \left\{1 - \left[1 - \omega_t \cdot r(\hat{\overline{\sigma}})\right]d_c\right\}\left\{1 - \left[1 - \omega_c + \omega_c \cdot r(\hat{\overline{\sigma}})\right]d_t\right\}, \ 0 \le \omega_t, \omega_c \le 1 \tag{3}$$

where $\omega_t$ and $\omega_c$ are the material properties that control the damage recovery upon load reversal. When $\omega_t$ takes the value 1, the material will fully recover the tensile stiffness while the load changes from pressure to tension regardless of the damage that has already occurred; and when it takes the value 0, it means the damage caused by the compression will affect the tensile stiffness completely. $\omega_c$ affects the compressive stiffness in the same way. $d_t$ and $d_c$ are the uniaxial damage variables to characterize the tensile damage and compressive damage, respectively. The value of 1 means the material is fully damaged and the value of 0 means the material is undamaged. $r(\hat{\overline{\sigma}})$ is a function of effective stress, defined as

$$r(\hat{\overline{\sigma}}) \overset{\text{def}}{=} \frac{\sum_{i=1}^3 \langle \hat{\overline{\sigma}}_i \rangle}{\sum_{i=1}^3 |\hat{\overline{\sigma}}_i|}, 0 \le r(\hat{\overline{\sigma}}) \le 1 \tag{4}$$

where $\hat{\overline{\sigma}}$ denotes a vector consisting of eigenvalues of the effective stress tensor, and $\hat{\overline{\sigma}}_i$ denotes its components. $\langle \cdot \rangle$ denotes the Macauley bracket, defined by $\langle x \rangle = (|x| + x)/2$.

The yield function of the model has the form

$$F(\overline{\sigma}) = \frac{1}{1-\alpha}\left(\overline{q} - 3\alpha\overline{p} + \beta\langle \hat{\overline{\sigma}}_{\max} \rangle - \gamma\langle -\hat{\overline{\sigma}}_{\max} \rangle\right) + \overline{\sigma}_c \le 0 \tag{5}$$

The parameters $\alpha$, $\beta$ and $\gamma$ in Equation (5) are given as

$$\alpha = \frac{\sigma_{b0} - \sigma_{c0}}{2\sigma_{b0} - \sigma_{c0}}, \; \beta = \frac{(\alpha - 1)\overline{\sigma}_c}{\overline{\sigma}_t} - (1 + \alpha), \; \gamma = \frac{3(1 - K_c)}{2K_c - 1}, \tag{6}$$

where $\overline{\sigma}_c$ and $\overline{\sigma}_t$ are parameters referred to as the effective uniaxial cohesion stresses, which have close relationships with the hardening variables of the model; $\overline{p}$ is the effective hydrostatic pressure, $\overline{p} = -\overline{\sigma}_{ii}/3$; $\overline{q}$ is the Mises equivalent effective stress, $\overline{q} = \sqrt{3(\overline{S}_{ij}\overline{S}_{ij})/2}$; $\overline{S}_{ij}$ is the component of deviatoric effective stress tensor, $\overline{S}_{ij} = \overline{p}\delta_{ij} + \overline{\sigma}_{ij}$, where $\delta_{ij}$ is the Kronecker delta; $\hat{\overline{\sigma}}_{max}$ is the algebraically maximum eigenvalue of effective stress tensor; $\sigma_{b0}$ and $\sigma_{c0}$ are the equibiaxial and uniaxial compressive yield stress of the material, respectively; $K_c = \overline{q}_{TM}/\overline{q}_{CM}$, where $\overline{q}_{TM}$ is the Mises equivalent effective stress corresponding to an arbitrary point on the projection trace of yield surface on the tensile meridian for any $\overline{p} < 0$ and $\overline{q}_{CM}$ corresponds to such a point on the compressive meridian. The relationship between $K_c$ and the projection of the yield surface on the deviatoric plane are shown in Figure 6a. The yield surface in the plane stress condition is shown in Figure 6b.

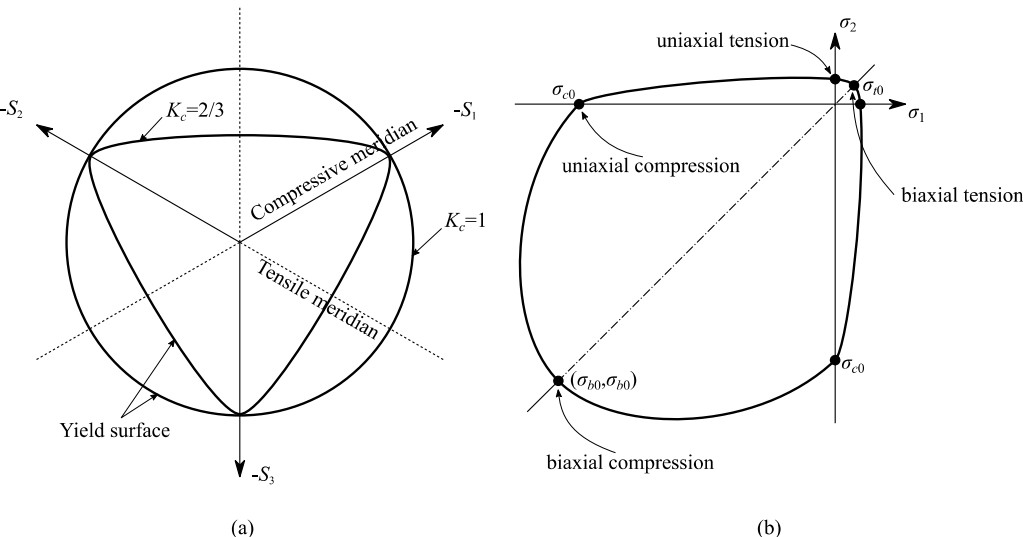

(a)

(b)

**Figure 6.** Schematic of the yield surface of the model. (**a**) Projection trace of yield surface on the deviatoric plane and (**b**) yield surface in the plane stress condition.

The model assumes a nonassociated potential flow, with the Drucker–Prager hyperbolic flow potential function:

$$G = \sqrt{(\kappa f_{t,r} \tan \psi)^2 + \overline{q}^2} - \overline{p} \tan \psi \tag{7}$$

where $f_{t,r}$ is the uniaxial tensile strength. $\kappa$ is a parameter referred to as the eccentricity, which defines the rate at which the function approaches the asymptote.

For a more detailed definition of the damaged plasticity model, please refer to the ABAQUS$^{TM}$ user manual.

The parameter calibration of the damaged plasticity model requires the uniaxial stress–strain curve of the material, which can be determined according to the following formula by referring to the Chinese Standard GB50010-2010 [22]:

For the uniaxial tension:

$$\sigma = \begin{cases} \frac{\varepsilon}{\varepsilon_{t,r}}\left[1.2f_{t,r} - 0.2f_{t,r}\left(\frac{\varepsilon}{\varepsilon_{t,r}}\right)^5\right], & \frac{\varepsilon}{\varepsilon_{t,r}} \leq 1 \\ \frac{\varepsilon}{\varepsilon_{t,r}}\dfrac{f_{t,r}}{\alpha_t\left(\frac{\varepsilon}{\varepsilon_{t,r}} - 1\right)^{1.7} + \frac{\varepsilon}{\varepsilon_{t,r}}}, & \frac{\varepsilon}{\varepsilon_{t,r}} > 1 \end{cases} \tag{8}$$

For the uniaxial compression:

$$\sigma = \begin{cases} \dfrac{\varepsilon}{\varepsilon_{c,r}} \dfrac{n f_{c,r}}{n-1+\left(\frac{\varepsilon}{\varepsilon_{c,r}}\right)^n}, & \dfrac{\varepsilon}{\varepsilon_{c,r}} \leq 1 \\[3mm] \dfrac{\varepsilon}{\varepsilon_{c,r}} \dfrac{f_{c,r}}{\alpha_c\left(\frac{\varepsilon}{\varepsilon_{c,r}}-1\right)^2 + \frac{\varepsilon}{\varepsilon_{c,r}}}, & \dfrac{\varepsilon}{\varepsilon_{c,r}} > 1 \end{cases} \tag{9}$$

where

$$n = \frac{E_0 \varepsilon_{c,r}}{E_0 \varepsilon_{c,r} - f_{c,r}} \tag{10}$$

where $f_{c,r}$ is the ultimate uniaxial compressive strength of concrete. $E_0$ is the elastic modulus of concrete. $\varepsilon_{t,r}$ is the tension strain corresponding to the tensile strength, and $\varepsilon_{c,r}$ is the compression strain corresponding to the ultimate compressive strength. $\alpha_t$ and $\alpha_c$ are shape coefficients related to the descending section of concrete uniaxial stress–strain curve, which can be referred via the standard.

The damage variables, $d_t$ and $d_c$, can be calculated according to the following steps. As shown in the Figure 7, for the uniaxial tension, after the material produces inelastic strain $\varepsilon_t^{in}$, due to the material elastic modulus that has been weakened, the unloading slope $(1-d_t)E_0$ will be less than the initial elastic slope $E_0$, this means that in the inelastic strain, only the part that will not restore during the unloading is the real plastic strain $\varepsilon_t^{pl}$. Assuming that the proportion of the recoverable part in the inelastic strain is $\eta_t$, the damage variable $d_t$ can be calculated as

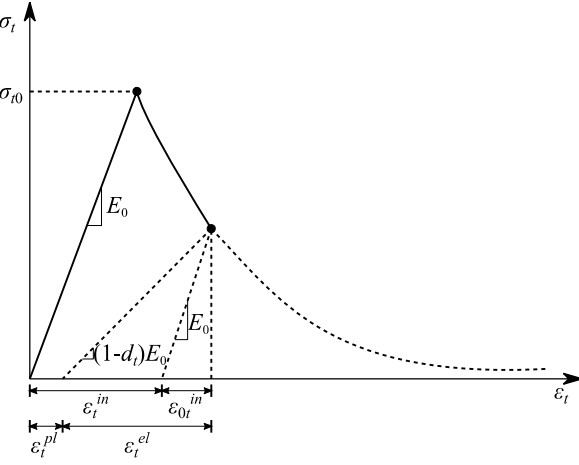

**Figure 7.** Relationship between the inelastic strain $\varepsilon_t^{in}$ and the plastic stain $\varepsilon_t^{pl}$ under uniaxial tension.

$$d_t(\varepsilon) = \frac{(1-\eta_t)E_0 \varepsilon_t^{in}(\varepsilon)}{\sigma_t(\varepsilon) + (1-\eta_t)E_0 \varepsilon_t^{in}(\varepsilon)} \tag{11}$$

where

$$\varepsilon_t^{in}(\varepsilon) = \varepsilon - \varepsilon_{t,r} \tag{12}$$

The value of $\eta_t$ should be obtained by the test. When there is no test data, according to the study in the literature [23], the value of $\eta_t$ typically ranging from 0.5 to 0.95 for concrete, generally 0.9 is acceptable.

Concrete under uniaxial compression has successively undergone an elastic deformation stage, a plastic hardening stage and a plastic softening stage. In the stress–strain curve, the stress value corresponding to the junction between the elastic deformation stage and the plastic hardening stage is called the proportional ultimate strength $f_{c,ppt}$, and this stress value is the true initial yield point in the

ABAQUS$^{\text{TM}}$ constitutive model. The proportional ultimate strength $f_{c,ppt}$ and the ultimate uniaxial compressive strength $f_{c,r}$ have the following empirical relationship:

$$f_{c,ppt} = 0.7 f_{c,r} \tag{13}$$

Substitute $f_{c,ppt}$ to Equation (9) to obtain the corresponding strain $\varepsilon_{c,ppt}$ (may need a trial calculation). Then the damage variable $d_c$ can be calculated as

$$d_c(\varepsilon) = \frac{(1 - \eta_c) E_0 \varepsilon_c^{in}(\varepsilon)}{\sigma_c(\varepsilon) + (1 - \eta_c) E_0 \varepsilon_c^{in}(\varepsilon)} \tag{14}$$

where

$$\varepsilon_c^{in}(\varepsilon) = \varepsilon - \varepsilon_{c,ppt} \tag{15}$$

In Equation (14), $\eta_c$ is the proportion of the recoverable part in the inelastic compression strain, typically has the value ranging from 0.35 to 0.7, generally 0.6 is acceptable.

Thus, the scalar stiffness degradation variable, $d$, can be obtained using Equation (3). The uniaxial stress–strain curves and the corresponding values of scalar stiffness degradation variable, of C30 shotcrete and C50 concrete, which were used as the material of the shotcrete layer and the core concrete respectively, are shown in Figure 8.

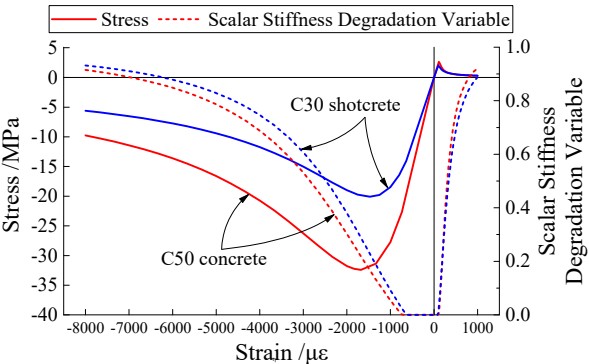

**Figure 8.** Diagram of the stress and scalar stiffness degradation variable.

As for the other model parameters, the eccentricity $\kappa$ has little influence on the calculation, it can simply take the value 0.1. For the biaxial compressive strength ratio $\sigma_{b0}/\sigma_{c0}$, many research results prove that the value of $\sigma_{b0}/\sigma_{c0}$ is generally stable at 1.10–1.16, which is generally 1.16. According to the definition, the tensile and compressive yield ratio $K_c$ is a function of stress invariance, but the existing test results show that $K_c$ is basically a certain value, and the general value is $K_c = 2/3$. For most of the quasi-brittle materials including concrete, the cracks caused by the tension process will be closed while the material transferred from tension to compression. However, on the other hand, the internal microcracks due to compression and shearing will not disappear while the material turned to be tensioned. Therefore, the stiffness recovery coefficients are generally taken as $\omega_t = 0$ and $\omega_c = 1$.

The concrete material parameters used in the calculation are shown in Table 1.

**Table 1.** Material parameters of concrete.

| Parameter | Description/Unit | Value | |
|---|---|---|---|
| | | **C50 Concrete** | **C30 Shotcrete** |
| $E_0$ | Initial elastic modulus/GPa | 34.5 | 25 |
| $\nu$ | Poisson's ratio | 0.2 | 0.2 |
| $\rho$ | Mass density/t·m$^{-3}$ | 2.3 | 2.2 |
| $f_{c,r}$ | Ultimate uniaxial compressive strength/MPa | 32.4 | 20.1 |
| $\varepsilon_{c,r}$ | Compression strain corresponding to $f_{c,r}$/µε | 1678 | 1471 |
| $\alpha_c$ | Shape coefficient of uniaxial compression $\sigma$-$\varepsilon$ curve | 1.499 | 0.746 |
| $f_{t,r}$ | uniaxial tensile strength/MPa | 2.64 | 2.01 |
| $\varepsilon_{t,r}$ | Tension strain corresponding to $f_{t,r}$/µε | 110.1 | 95.24 |
| $\alpha_t$ | Shape coefficient of uniaxial tension $\sigma$-$\varepsilon$ curve | 2.191 | 1.264 |
| $\psi$ | Dilation angle/° | 15 | 15 |
| $\sigma_{b0}/\sigma_{c0}$ | Biaxial compressive strength ratio | 1.16 | 1.16 |
| $K_c$ | Tensile and compressive yield ratio | 0.67 | 0.67 |
| $\kappa$ | Eccentricity of plastic flow potential surface | 0.1 | 0.1 |
| $\omega_c$ | Stiffness recovery coefficients of tension–compression | 1 | 1 |
| $\omega_t$ | Stiffness recovery coefficients of compression–tension | 0 | 0 |

### 3.2.2. Steel

The steel material adopts the metal plasticity model in ABAQUS$^{TM}$. The yield function has the form:

$$F = q - \sigma^0 \leq 0 \tag{16}$$

where $q$ is the Mises equivalent stress, $q = \sqrt{3J_2} = \sqrt{3S_{ij}S_{ij}/2}$, and $S_{ij}$ is the component of the deviatoric stress tensor. $\sigma^0$ is the yield stress. When hardening of the material is taken into account, the yield stress is a function of the hardening parameter.

The model assumes an associated potential flow, and adopts the isotropic hardening rule. The hardening parameter is the equivalent plastic strain $\widetilde{\varepsilon}^{pl}$, defined as

$$\widetilde{\varepsilon}^{pl} = \int \sqrt{d\varepsilon_{kl}^{pl} d\varepsilon_{kl}^{pl}} \tag{17}$$

When hardening is considered, the stress–strain curve of the material needs to be provided as the model input. The uniaxial tensile stress–strain curve of common low-carbon steel can be simplified as a five-segment piecewise function expressed as [24]:

$$\sigma = \begin{cases} E_S\varepsilon, & \varepsilon < \varepsilon_e \\ -\frac{E_S}{\varepsilon_e}\varepsilon^2 + 3E_S\varepsilon - 0.8f_y, & \varepsilon_e \leq \varepsilon < \varepsilon_{e1} \\ f_y, & \varepsilon_{e1} \leq \varepsilon < \varepsilon_{e2} \\ f_y\left[1 + \left(\frac{f_u}{f_y} - 1\right)\frac{\varepsilon - \varepsilon_{e2}}{\varepsilon_{e3} - \varepsilon_{e2}}\right], & \varepsilon_{e2} \leq \varepsilon < \varepsilon_{e3} \\ f_u, & \varepsilon_{e3} \leq \varepsilon \end{cases} \tag{18}$$

where $E_S$ is the Young's modulus of steel. $f_y$ and $f_u$ are the yield stress and ultimate strength of steel, respectively. $\varepsilon_e$, $\varepsilon_{e1}$, $\varepsilon_{e2}$, and $\varepsilon_{e3}$ are the strain measures corresponding to the four junction point of the following five stages of deformation: linear deformation stage, elastoplastic deformation stage, plastic flow stage, strengthening stage, and necking stage. A rule-of-thumb is that set $\varepsilon_e = 0.8f_y/E_s$, $\varepsilon_{e1} = 1.5\varepsilon_e$, $\varepsilon_{e2} = 10\varepsilon_{e1}$, and $\varepsilon_{e3} = 100\varepsilon_{e1}$.

By referring to the Chinese Standard GB50017-2017 [25] and literature [24], the material parameters used in the calculation are listed in Table 2.

**Table 2.** Material parameters of steel.

| Parameter | Description/Unit | Value |
|:---:|:---:|:---:|
| $E_S$ | Young's modulus/GPa | 206 |
| $\nu$ | Poisson's ratio | 0.31 |
| $\rho$ | Mass density/t·m$^{-3}$ | 7.85 |
| $f_y$ | Yield stress/MPa | 279 |
| $f_u$ | Ultimate strength/MPa | 450 |

*3.3. Research Cases*

In terms of the size of CFST, two research groups were set up, based on the benchmark of two aspects, respectively:

1. Group 1: With the benchmark of steel consumption

Steel consumption is one of the main economic indicators of the support. In the case where other parameters are the same, the cross-sectional area of the steel is proportional to the steel consumption. At present, H175 profile steel is often used as the rigid frame in Chinese traffic tunnels. The cross-sectional area of the H175 profile steel was 5142 mm$^2$, and we used this as the benchmark, the research cases in group 1 were formulated as shown in Table 3. Note that as for the rectangular-shaped and trapezoidal-shaped CFST section, the steel tube width $w_S$ was set to be equal to the steel tube height $h_S$. The interior angle $\alpha$ was set to 60° for the triangular-shaped CFST and 75° for the trapezoidal-shaped CFST, respectively.

2. Group 2: With the benchmark of steel tube height (or outer diameter)

Due to the poor tensile strength of concrete, the bending capacity of the steel frame in the support is very important. In order to obtain a higher bending modulus, the height of the steel frame should be raised as much as possible. In this research group, the section height of the I22a profile steel, which is another type of rigid frame commonly used in the Chinese traffic tunnel, was used as the benchmark of the steel tube height (or steel tube outer diameter). The wall thickness of the steel tube was uniformly set to 10 mm. Table 4 shows the sectional parameters of research cases in the current research group. The choice of steel tube width $w_S$ and interior angle $\alpha$ were the same as that of research group 1.

**Table 3.** Cross-sectional parameters of research cases in group 1.

| Cross-Sectional Form | Steel Tube Height/Diameter (mm) | Wall Thickness (mm) | Case Alias | Cross-Sectional Area of Steel (mm$^2$) | Axial Compressive Modulus (GN) | Bending Modulus (MN·m$^2$) |
|---|---|---|---|---|---|---|
| Circular | 213 | 8 | Cir-D213×8 | 5152 | 2.91 | 15.13 |
| Rectangular | 169 | 8 | Squ-H169×8 | 5152 | 2.84 | 13.99 |
| Triangular | 220 | 12 | Tri-H220×12 | 5031 | 2.82 | 16.92 |
| Trapezoidal | 195 | 8 | Tra-H195×8 | 5148 | 2.84 | 15.1 |
| H175 profile steel | 175 | - | H175 | 5142 | 2.62 | 14.77 |

Note: 'Axial Compressive modulus' is defined as $\Sigma E_i \times A_i$, and 'Bending modulus' is defined as $\Sigma E_i \times I_i$, where $E_i$, $A_i$, and $I_i$ are the young's modulus, cross-sectional area, and moments of inertia of $i$th component of the member (steel, concrete, shotcrete), respectively.

**Table 4.** Cross-sectional parameters of research cases in group 2.

| Cross-Sectional Form | Steel Tube Height/Diameter (mm) | Wall Thickness (mm) | Case Alias | Cross-Sectional Area of Steel (mm$^2$) | Axial Compressive Modulus (GN) | Bending Modulus (MN·m$^2$) |
|---|---|---|---|---|---|---|
| Circular | 220 | 10 | Cir-D220×10 | 6597 | 3.18 | 16.86 |
| Rectangular | 220 | 10 | Squ-H220×10 | 8400 | 3.59 | 21.99 |
| Triangular | 220 | 10 | Tri-H220×10 | 4227 | 2.68 | 16 |
| Trapezoidal | 220 | 10 | Tra-H220×10 | 7221 | 3.27 | 19.26 |
| I22a profile steel | 220 | - | I22a | 4213 | 2.45 | 15.67 |

## 4. Results and Analysis

### 4.1. Analysis of Research Group 1

Figure 9 shows the load–crown settlement curve of the research group 1 with the benchmark of steel consumption. It can be seen that the shape of the load–crown settlement curve was similar in all cases. When the load reached about 900 kPa, the stiffness of each case began to differentiate. Among them, the stiffness of the case of Tri-H220×12 was obviously greater than that of other cases, which may caused by the larger cross-sectional height and wall thickness of the steel tube. Each case reached its ultimate capacity when the crown settlement reached about 28 mm. The ultimate capacity of the case of Tri-H220×12 was significantly higher, which was 1383.18 kPa. The rest of the cases had similar ultimate capacities of about 1100 kPa.

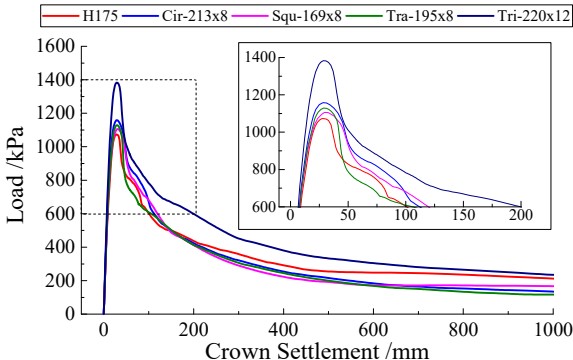

**Figure 9.** Load–crown settlement curve of the members in research group 1.

After each case reached the ultimate capacity, as the load further increased, the load–crown settlement curve quickly turned to decline, and there was no obvious step that appeared. This shows that the member will quickly enter the buckling state when it reaches the ultimate capacity. After the crown settlement exceeded 200 mm in each case, the decrease rate of the curve tended to be stable. At this time, the bearing capacity of each member dropped to less than half of the ultimate capacity. The bearing capacity that the case of Tri-H220×12 could maintain was slightly higher than that of other cases in the post-buckling stage.

Figure 10a–d is the deformation curve, the curvature distribution curve, and the hoop strain distribution curve at the outer and inner surface of the member in order. These curves are based on the values at the end of the analysis (when crown settlement reaches 2 m) of research group 1. It can be seen from the figure that after the circular arch member buckled under the normal uniform load, the bending mainly occurred in two regions: the crown of the arch was undergoing inward bending, and the spandrel of the arch underwent outward bending. The curvature at the crown was slightly larger than that at the spandrel. Compared with the initial value, the curvature from the spandrel to the arch spring was slightly increased, but the degree of bending was not large. The original curvature was basically maintained in a part of the area above the spandrel. Due to the bending of the spandrel, the member axis in the area above the spandrel was severely offset from the initial position.

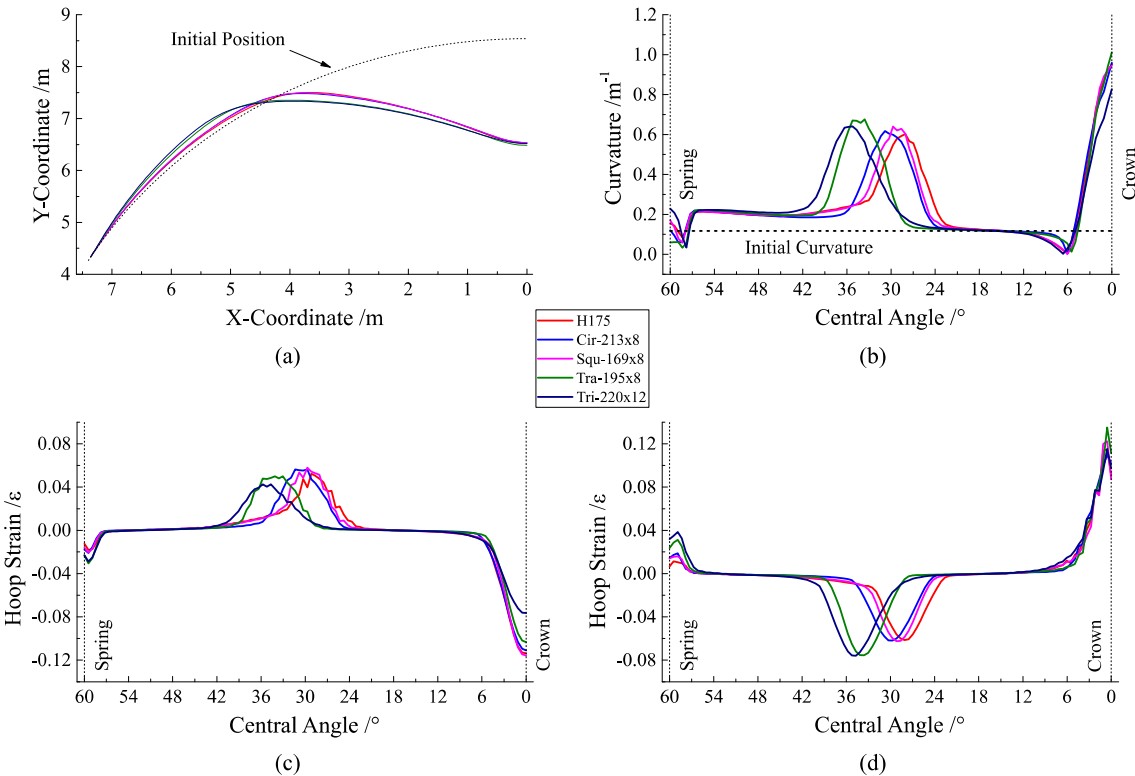

**Figure 10.** Deformation characteristics diagram of the members in research group 1. (**a**) Deformation curve; (**b**) curvature; (**c**) hoop strain at the outer surface; and (**d**) hoop strain at the inner surface.

The deformation modes of each case could be roughly divided into two categories. The deformation modes of the case of Cir-D213×8, case of Squ-H169×8, and case of H175 after buckling were similar. The maximum curvature of the spandrel appears near the central angle of 30°, and the absolute values of the hoop strain at the outer and the inner surface were close. The case of Tri-H220×12 and case of Tra-H195×8 had the maximum curvature appear near the central angle of 36°, and the absolute value of the hoop strain at the inner surface was significantly greater than that at the outer surface. It can be seen from the deformation curve that the case of Tri-H220×12 and case of Tra-H195×8 had a larger horizontal offset than the other cases near the spandrel, which may be caused by the larger bending degree at the arch spring of these two cases. This phenomenon is presumably caused by the unsymmetrical cross-sectional form of the case of Tri-H220×12 and case of Tra-H195×8 along its horizontal axis, resulting in their ability to resist inward bending to be weaker than the ability to resist outward bending.

When the arch member works in the elastic state, if the external load distribution form is close to the shape of the arch axis, the member mainly bears the axial force and only a small bending moment will take place, theoretically. On the other hand, in the elastic state, the internal force is only related to its moment of inertia and elastic modulus, and it cannot reflect the difference between various cross-sectional forms. Therefore, it is meaningless to analyze the internal force in the elastic state. After the member enters the buckling state, the shotcrete in the tension zone can hardly continue to withstand the tensile force, and the shotcrete in the compression zone will also be crushed to some extent, which makes the balance of the internal force greatly changed. In this circumstance, the ideal state is that the CFST frame takes over from the shotcrete layer to continuously work by bearing most of the tensile force and bending moment. Therefore, by observing the internal force distribution in the buckling state, the cooperative performance of shotcrete, steel tube, and core concrete can be evaluated to judge the advantages and disadvantages of each cross-sectional form.

Figure 11 is the internal force distribution diagram of each case at the end of the analysis. Corresponding to their bending modes, the total bending moment distribution of each member was roughly the same. The maximum positive bending moment (caused by outward bending) appeared in the range of the central angle of 30–40°, and the maximum negative bending moment occurred in the crown; the total axial force of each case was almost the compressive force and the distribution was relatively uniform.

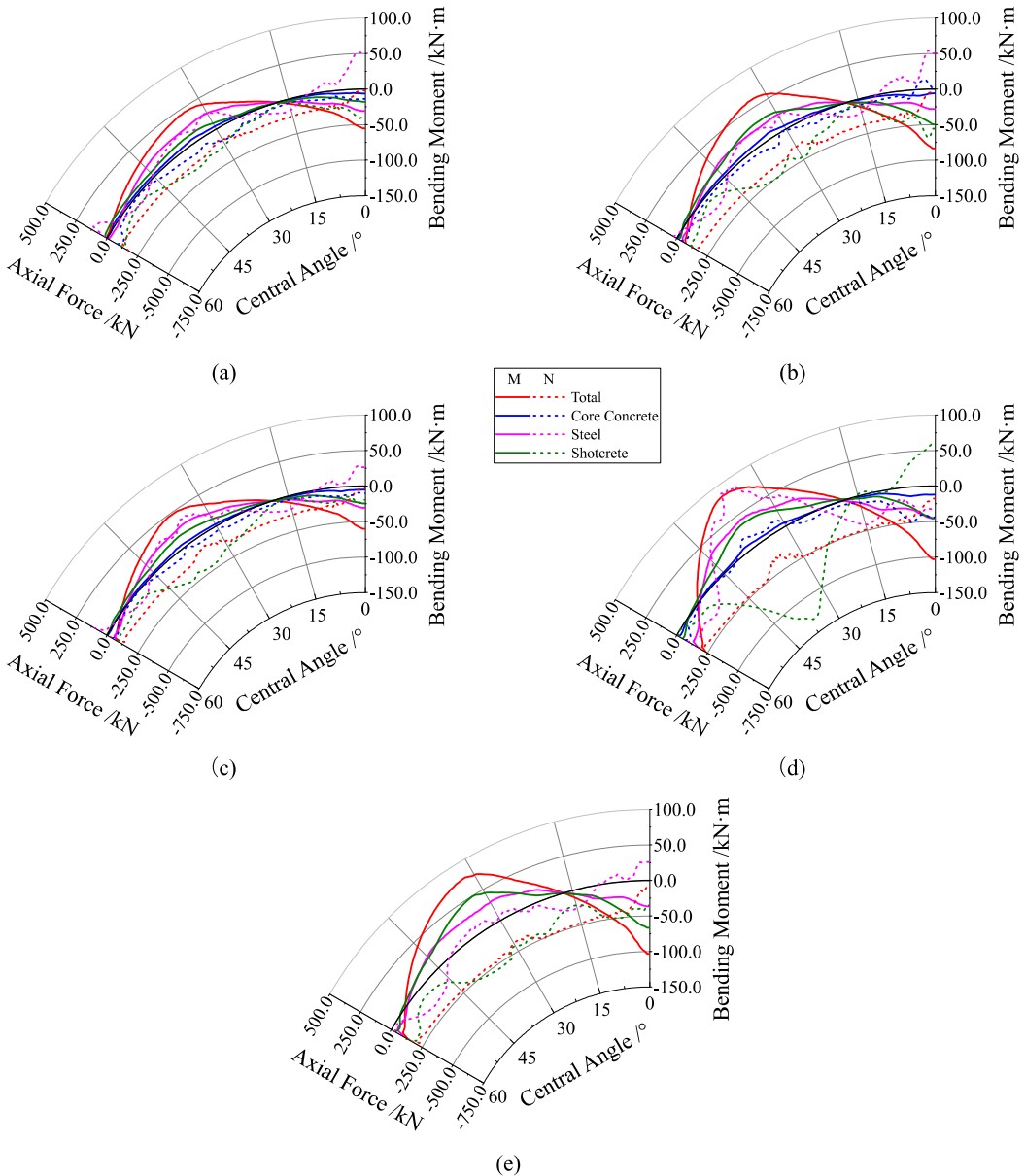

**Figure 11.** Internal force distribution diagram of the members in research group 1. (**a**) Case of Cir-213×8; (**b**) case of Squ-169×8; (**c**) case of Tra-195×8; (**d**) case of Tri-220×12; and (**e**) case of H175.

The bending moment of the case of Cir-D213×8 was mostly borne by the steel tube, and the core concrete and shotcrete hardly bore the bending moment; the axial force of the steel tube at the spandrel and the crown was the tensile force, and the core concrete and shotcrete bore the compressive force. Overall, for the case of Cir-D213×8, the core concrete, the steel tube, and the shotcrete had a relatively reasonable proportion of internal force distribution. That means all components of the member fully exert their own functions, and the collaborative performance was good.

The bending moment of the case of Squ-H169×8 was mainly shared by the steel tube and shotcrete, the share proportion was nearly the same. For the case of Squ-H169×8, the bending moment and axial force of the shotcrete were larger than the case of Cir-D213×8, which was due to the smaller cross-sectional size of the steel tube and the larger area of the shotcrete. The axial force and bending moment undertaken by the core concrete of the case of Squ-H169×8 were both small, which means the core concrete had only a small contribution to the bearing performance. This may be due to the lack of ability of the square-shaped steel tube to constrain the core concrete. Overall, the collaborative performance of the case of Squ-H169×8 was relatively poor.

The internal force distribution mode of the case of Tri-H220×12 was the most special. Like the case of Squ-H169×8, the bending moment was mainly shared by the steel tube and shotcrete equally. In the region near the spandrel, the steel tube bore a higher tensile force, and the shotcrete bore a higher compressive force, which is an ideal situation. In addition, the core concrete in this region also bore a part of the tensile force, which shows that the triangular steel tube had a strong constraint on the core concrete, improving the tensile strength of the core concrete. In the region near the crown, the case of Tri-H220×12 shows a completely different state of stress. The shotcrete bore a higher tensile force, and the steel tube and core concrete were compressed. This phenomenon may be related to the obvious unsymmetrical cross-sectional form of the case of Tri-H220×12. Overall, for the case of Tri-H220×12, the various components of the member could form a complementary effect when they are bent, and the performance was excellent, especially when they resist outward bending.

The internal force distribution of the case of Tra-195×8 was similar to the case of Squ-H169×8, but the bending moment and axial force of the shotcrete and core concrete were significantly lower. It can be considered that it was difficult to effectively mobilize the strength of the shotcrete and core concrete. At the most strongly deformed part of the H175 member, its bending moment was mainly borne by the shotcrete. The bending moment of the shotcrete was about twice that of the profile steel. It can be considered that, for the case of H175, the profile steel in the member contributed relatively little to the overall bearing capacity, but the opposite was that the shotcrete could play its performance better.

In summary, from the perspective of the bearing capacity of the member, under the same steel consumption, the triangular-shaped CFST member has obvious advantages. The bearing capacity of the triangular-shaped CFST was higher than the others. However, at the same time, due to the unequal bidirectional bending resistance of the triangular-shaped CFST member, the distribution of the hoop strain in its cross-section was asymmetric, and its deformation mode was slightly different from that of members with equal bidirectional bending resistance. From the perspective of the internal force distribution, the cooperative performance of the triangular-shaped CFST member was the most ideal. The circular-shaped CFST was next. The rectangular-shaped and trapezoidal-shaped CFST could not fully mobilize the capabilities of the core concrete, therefore, their cooperative performance was relatively poor.

*4.2. Analysis of Research Group 2*

The load–crown settlement curve of the research group 2 with the benchmark of the steel tube height is shown in Figure 12. Under the same steel tube height and steel tube wall thickness, the case of Squ-H220×10 had the highest ultimate bearing capacity of 1567.74 kPa, the case of Tra-H220×10 was followed by 1401.55 kPa, the case of Cir-D220×10 was again 1361.56 kPa, and the case of Tri-H220×10 had the lowest ultimate bearing capacity of 1261.55 kPa among all of the cases using the CFST frame. Additionally, the ultimate bearing capacity of the case of I22a was only 1002.56 kPa, which was significantly lower than that of the CFST cases. However, it can be seen from Table 4 that this sequence was the same as the order of the cross-sectional area of the steel sections from large to small. Thus, there is a clear positive correlation between the cross-sectional area of the steel and the ultimate bearing capacity.

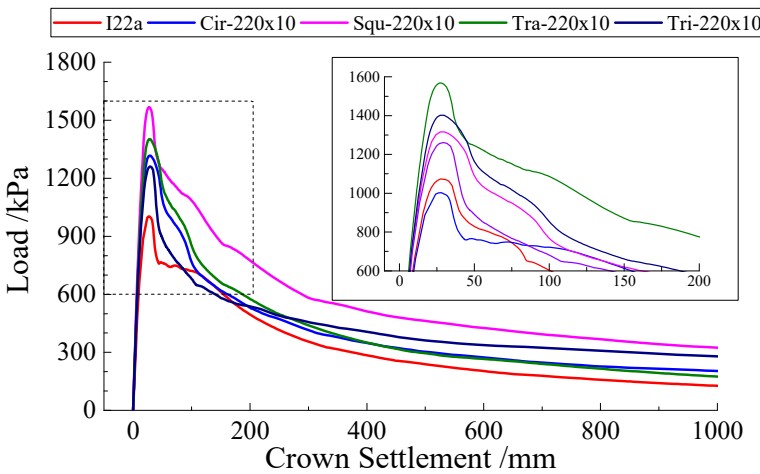

**Figure 12.** Load–crown settlement curve of the members in research group 2.

Although the ultimate bearing capacity of the case of Tri-H220×10 was relatively lower, after the member buckled, the bearing capacity of the member gradually surpassed the other cases except the case of Squ-H220×10 with the development of a deformation. It can be considered that the load-bearing capacity of the triangular-shaped CFST member after buckling was relatively strong. Generally speaking, it is clear that the bearing capacity was proportional to the cross-sectional area of steel. However, taking the case of Squ-H220×10 and Tri-H220×10 as examples, the former had a steel cross-sectional area of 8400 mm$^2$, nearly twice as the latter, which the area was 4226.79 mm$^2$. However, the ultimate bearing capacity of the former was only 124.27% of the latter, and the increase in the bearing capacity was not proportional to the increase in steel consumption. Therefore, although the use of rectangular-shaped and trapezoidal-shaped CFST could obtain a higher bearing capacity, its economy was far less than that of circular-shaped and triangular-shaped CFST.

Figure 13a–d is the deformation curve, the curvature distribution curve, and the hoop strain distribution curve at the outer and inner surface of the member in order, with the values obtained at the end of the analysis of research group 2. Comparing Figure 10, it is not difficult to find that in the research group 1, the deformation mode of the trapezoidal-shaped CFST member was close to that of the triangular-shaped CFST, which shows a significant difference in the bidirectional bending resistance. However, in this group, the deformation mode of the trapezoidal-shaped CFST member was obviously close to that of circular-shaped and square-shaped CFST members, which had the maximum bending region appear near the central angle of 30°. This shows that with the increase of the height of the steel tube, the difference of the bidirectional bending resistance of the trapezoidal-shaped CFST member decreased.

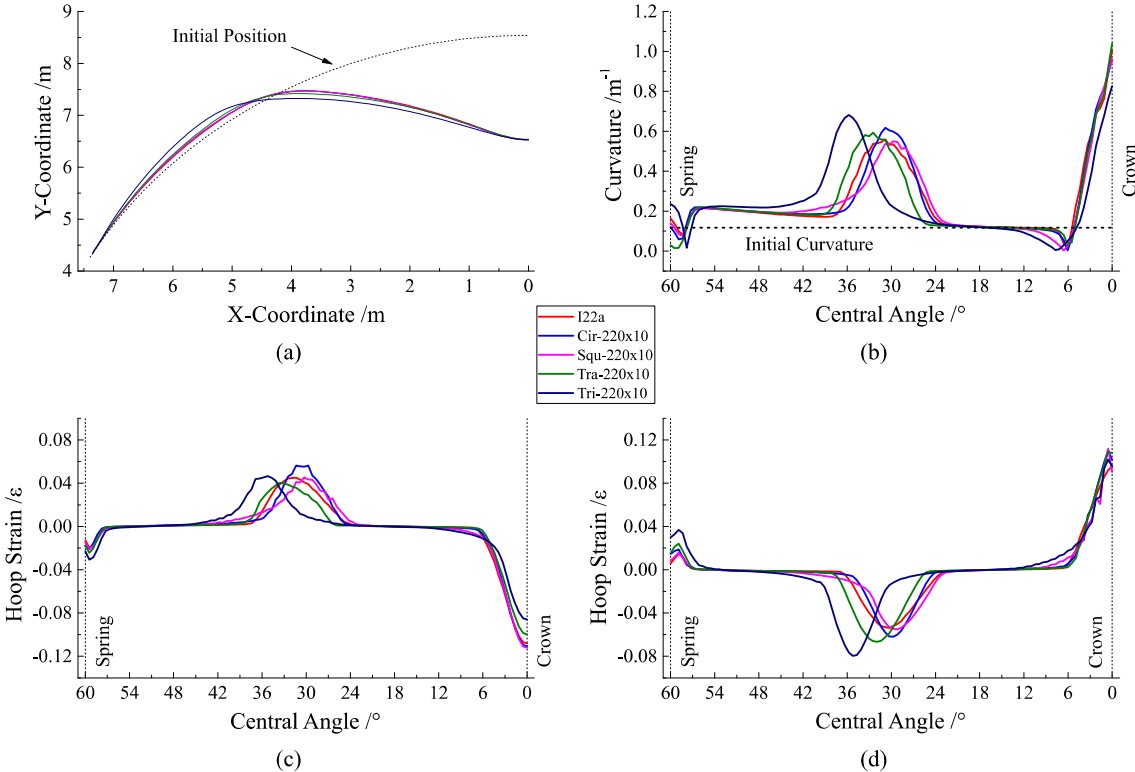

**Figure 13.** Deformation characteristics diagram of the members in research group 2. (**a**) Deformation curve; (**b**) curvature; (**c**) hoop strain at outer surface; and (**d**) hoop strain at inner surface.

Figure 14 is the internal force distribution diagram of the cases in the research group 2. It can be seen that as the height and wall thickness of the steel tube increased, the bearing proportion of the bending moment increased, for the steel tube and the core concrete. This is especially obvious for the rectangular-shaped and trapezoidal-shaped CFST members. It can be inferred that the proportion of the cross-sectional area of the steel tube, core concrete, and shotcrete in the entire cross-section had a great influence on the stress state of the member. In addition, the larger height of the steel tube was conducive to the performance of the CFST frame. Comparing the cases in this group, it could be found that under the same steel tube height and wall thickness, the internal force bearing proportion and distribution law of steel tubes in circular-shaped and rectangular-shaped members were similar. Due to the large area of the core concrete in the rectangular-shaped CFST member, the internal force it bears was close to or even exceeded that of shotcrete. It can be found that when the height and wall thickness of the steel tube were sufficiently large, the rectangular-shaped CFST began to show its superiority. Its bearing capacity and cooperative performance were both excellent.

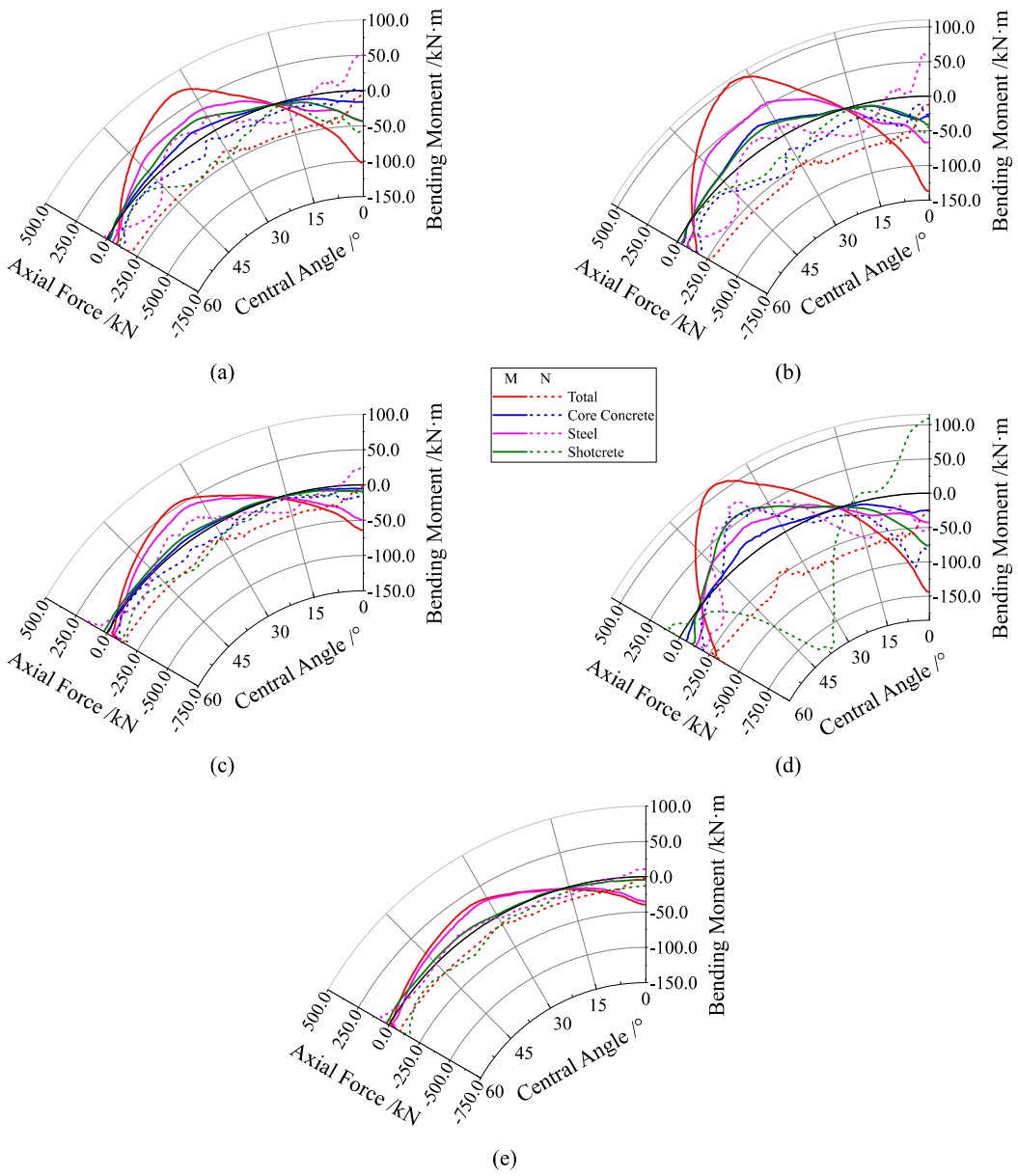

**Figure 14.** Internal force distribution diagram of the members in research group 2. (**a**) Case of Cir-220×10; (**b**) case of Squ-220×10; (**c**) case of Tra-220×10; (**d**) case of Tri-220×10; and (**e**) case of I22a.

The internal force distribution pattern of the case of Tri-220×10 was the same as the case of Tri-220×12 in the previous research group, but the reduction of wall thickness made the proportion of the internal force born by the steel tube to be significantly reduced. At the region near the spandrel and the crown, shotcrete bore a larger proportion of bending moment and axial force, which is more disadvantageous. In addition, what is obviously different from the case of Tri-220×12 is that the core concrete of the case of Tri-220×10 bore more axial force, which is also related to the reduction of the wall thickness of the steel tube. Therefore, when using triangular-shaped CFST, the wall thickness of the steel tube should be appropriately increased to ensure that the steel tube can work efficiently. In the case of Tra-220×10, the steel tube was the main load-bearing component. The core concrete and shotcrete only bore little bending moment and axial force. This further validates the conclusion that the trapezoidal-shaped CFST has poor cooperative performance.

### 4.3. Comprehensive Analysis and Suggestions on the Cross-Sectional Form

The calculation results of two research groups are now summarized. The following indicators were picked or defined to perform an overall analysis of the performance of the CFST members under various cross-sectional forms:

- Ultimate bearing capacity: the load corresponding to the peak point of the load–crown settlement curve.
- Residual bearing capacity: from the load–crown settlement curve, it can be seen that after the crown settlement reached 200 mm, the decrease rate of the curve tended to be stable. Therefore, the corresponding load on the curve when the crown settlement is 200 mm is defined as the residual bearing capacity.
- Mean effective stiffness: the ratio of the ultimate bearing capacity and the crown settlement corresponding to it. The mean effective stiffness can be a measure of the ability to resist the deformation without failure.
- Elastic stiffness: the ratio of the load and crown settlement corresponding to the critical point that the deformation mechanism of the member changes from elastic to elastoplastic. It can be a measure of the ideal resistance to deformation.

Figure 15 is the scatterplot of the performance indicators, which has taken the cross-sectional area of steel as the horizontal axis. As can be seen from Figure 15a, except for the triangular-shaped CFST members, the ultimate bearing capacity of the other members were basically linearly related to the steel cross-sectional area. When the cross-sectional area of steel was increased by 1000 mm$^2$, the ultimate bearing capacity of the member was increased by about 135 kPa. Under the premise of the same steel cross-sectional area, the ultimate bearing capacity of the triangular-shaped CFST members was significantly higher than that of other CFST members with a different shape of the steel tube. However, for the triangular-shaped CFST members, the increase in the ultimate bearing capacity of the unit steel cross-sectional area was basically the same as other members with different cross-sectional forms. Figure 15b shows that there was also a linear relationship between the residual bearing capacity and the cross-sectional area of the steel. As with the law of the ultimate bearing capacity, the residual bearing capacity of the two cases of triangular-shaped CFST was higher than this average regression line. In addition, the case of I22a and Squ-220×12 had a better cost–performance ratio of the residual bearing capacity than the rest.

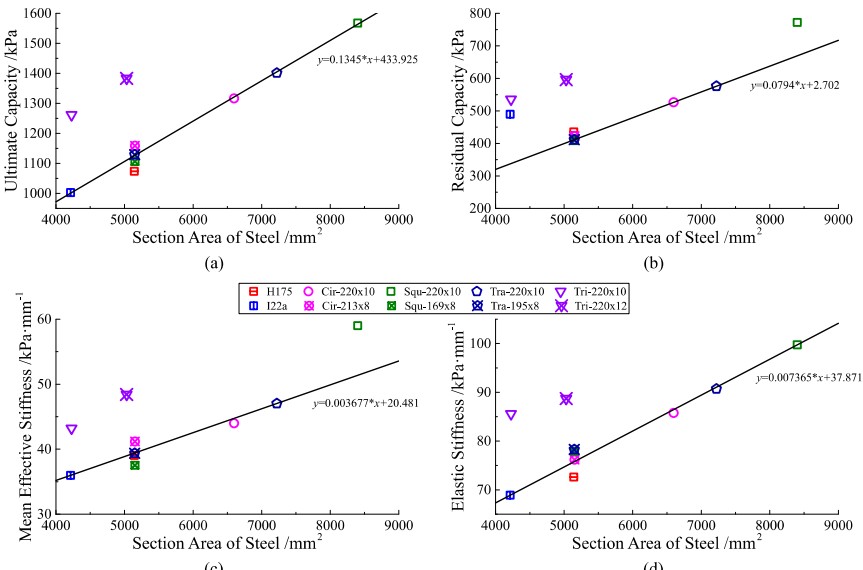

**Figure 15.** Scatterplot of the performance indicators summarized by both groups. (**a**) Ultimate capacity; (**b**) residual capacity; (**c**) mean effective stiffness; and (**d**) elastic stiffness.

Figure 15c,d clearly shows the similar linear relationship in terms of the mean effective stiffness and elastic stiffness. The stiffness of the triangular-shaped CFST members was significantly higher than other members with the same steel cross-sectional area. In terms of mean effective stiffness, in addition to the case of Tri-H220×12 and Tri-H220×10, the case of Squ-220×10 also exhibited a performance far higher than the average linear relationship, which is due to its extreme strong bending resistance. It can be assumed that the performance of the rectangular-shaped CFST member was greatly affected by the height of the steel tube. The stiffness and bearing capacity of the rectangular-shaped CFST can be very good when the height of steel tube is high enough. Thus, the rectangular-shaped CFST is suitable for the extreme case where the bearing capacity and stiffness are preferentially pursued.

Based on the above analysis, it could be found that the triangular-shaped CFST had the best bearing performance, especially after considering its high utilization rate of steel. The circular-shaped CFST had a more average performance in all aspects. Although the rectangular-shaped and trapezoidal-shaped CFST were not inferior in performance, they only had certain advantages when the size of the steel tube was large. Considering its low steel utilization rate and poor cooperative performance, in general, these two cross-sectional forms are not worth recommending.

## 5. Conclusions

The results of the research on the circular arch member abstracted from the supporting structure show that the closed-type CFST support had a higher ultimate bearing capacity and higher stiffness than the traditional tunnel support. The cross-sectional form had a great influence on the bearing performance and economic performance of the closed-type CFST support. Among the four types of closed-type CFST supports studied in this paper, the triangular-shaped CFST support shows the most excellent performance in all aspects. Under the premise of the same steel consumption, the bearing capacity and stiffness of the triangular-shaped CFST support were significantly higher than those of other CFST supports with different cross-sectional forms. Moreover, the triangular-shaped CFST support could maintain a high residual bearing capacity after the buckling. The triangular-shaped CFST support had a unique internal force distribution law, which proves the cooperative performance of its components was better. Considering that the triangular-shaped steel tube can theoretically satisfy the requirements of a close-fit with surrounding rock and ensure the compactness of shotcrete, it should be the preferred cross-sectional form for closed-type CFST support. However, the triangular-shaped CFST support has different bending resistance in different directions, and the impact of this feature should be fully considered when evaluating its bearing performance.

The overall performance of circular-shaped CFST support was moderate, but theoretically the circular-shaped steel tube can provide the most uniform constraining force to the core concrete, so its cooperative performance with the core concrete is better. In addition, the manufacturing process of the circular-shaped steel tube is the most mature, the material is easy to obtain, and the quality of core concrete pouring is easy to guarantee. Thus, circular-shaped CFST can also be an optional cross-sectional form of the closed-type CFST support. However, it is necessary to pay attention to the compactness of the shotcrete sprayed behind the steel tube when using the circular-shaped CFST. As for the rectangular-shaped and the trapezoidal-shaped CFST support, although they could meet the requirements in terms of the bearing capacity, their performance under the same steel consumption was slightly inferior to that of the circular-shaped CFST support. Additionally, under these two cross-sectional forms, the performance of the core concrete was insufficient, unable to take the unique advantages of CFST. The rectangular-shaped CFST support could only play a higher performance when the height of the steel tube is large, but this will inevitably lead to a substantial increase in the steel consumption. The trapezoidal-shaped CFST support had no obvious advantages in performance, and the processing difficulty was relatively high, so it is not recommended. The rectangular-shaped CFST support had acceptable performance, but the economy was the worst. It is recommended to only extreme conditions that the high bearing capacity and stiffness have the highest priority.

## 6. Discussion

This paper made some preliminary conclusions about the advantages and disadvantages of various cross-sectional forms of closed-type CFST support, but it should be noted that the object of calculation and analysis in this paper was only a partial abstraction in the tunnel support. In fact, the load distribution of the tunnel support was often not so ideal, and the support structure as a whole did not absolutely work in the form of an arch structure. This makes more in-depth research needed when applying the conclusions of this article to the design of the actual tunnel support structure. For example, for the triangular-shaped CFST support showing excellent performance in this study, when it was under the load of a complex situation, will its relatively weak inward bending resistance become its performance shortcomings or even become the weakest link that determines whether the structure is unstable? These are not yet known.

The design concept of the closed-type CFST support is to use the high tensile properties of steel to compromise the weakness of the tensile performance of the shotcrete. The restraining effect of the steel tube may increase the compressive capacity of the core concrete, thereby improving the bearing performance of the support structure under the compression-bending effect. Theoretically, any supporting structure conforming to this design concept can be treated as a similar structure. Therefore, for the support structure proposed in this paper, there are still many topics that can be studied, such as whether there is a more reasonable cross-sectional form other than the four forms studied in this paper, and whether it is feasible to replace the steel material with other organic polymer materials with high tensile properties. These questions are worth further discussion.

**Author Contributions:** Conceptualization, L.L.; Data curation, K.L.; Funding acquisition, L.L.; Investigation, K.L.; Project administration, L.L.; Software, K.L.; Visualization, K.L.; Writing—original draft, K.L.; Writing—review and editing, L.L. All authors have read and agreed to the published version of the manuscript.

**Funding:** The authors acknowledge the financial support provided by the Special Fund Project for Basic Research Expenses of Central Public Welfare Institutes (Grant No. 2018-9020).

**Conflicts of Interest:** The authors declare no conflict of interest.

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
