# Peer review of "Preliminary Design and Cross-Sectional Form Study of Closed-Type Concrete-Filled Steel Tube Support for Traffic Tunnel"

_symmetry, doi:10.3390/sym12081368_

Round 1

Reviewer 1 Report

The manuscript deals with a preliminary assessment of the performance of different steel profiles as supports for traffic tunnel sections. The authors are urged to consider the following remarks in a revised submission of the manuscript:

  1. The literature review needs to be enhanced. Refs. 1-14 & 17-20 are in Chinese. Hence, it is impossible for non-speakers to check the works cited. Also, the literature it abundant of experimental and theoretical works on CFSTs in English by both Chinese and non-Chinese researchers. The authors need to ensure that the literature is not biased and covers the state of the art in the field.
  2. It is not clear how the steel sections were initially selected. Were they selected so that e.g. they would have the same stiffness? Judging from the dimensions, four sections seem to have the same breadth/radius and thickness, but if this is the case, then by definition, all of them have different characteristics. It would be helpful to add a table with the sections’ geometrical properties.
  3. Based on the above remark, if the sections do not have similar properties, how are they assessed on a fair basis?
  4. 14 is not easy to visualize. Perhaps using thicker lines for all elements would solve this.
  5. An overall check for standard English and clarity of expression is needed. At some points it is hard to understand what the authors mean to state, or can be misleading. For example, based on the manuscript, it seems that the authors meant to say “optimal” instead of “optional” in the introduction.

Author Response

Thank you very much for your attention and comments on our paper. We have revised the manuscript according to your kind advices. Our responses are as follows.

Comment 1: The literature review needs to be enhanced. Refs. 1-14 & 17-20 are in Chinese. Hence, it is impossible for non-speakers to check the works cited. Also, the literature it abundant of experimental and theoretical works on CFSTs in English by both Chinese and non-Chinese researchers. The authors need to ensure that the literature is not biased and covers the state of the art in the field.

Response: Thanks for your criticism and suggestions. We have made a significant update to the literature review chapter. The research results of researchers from various countries have been summarized from the perspectives of theoretical analysis, numerical calculation, engineering applications, etc., which could supports the research of CFST support for traffic tunnels in this article.

Comment 2: It is not clear how the steel sections were initially selected. Were they selected so that e.g. they would have the same stiffness? Judging from the dimensions, four sections seem to have the same breadth/radius and thickness, but if this is the case, then by definition, all of them have different characteristics. It would be helpful to add a table with the sections’ geometrical properties.

Response: Thank you for your kind advice. The choice of the research plan is not based solely on the same bending or compressive stiffness. The original intention of the closed-type CFST support is to solve the problem of primary support failure, which often occurs in severe environments such as high ground stress/weak surrounding rock/special surrounding rock (such as expansive rock). Under this condition, it is difficult for us to ensure that the support can still be maintained in an elastic working state. Thus, I think it is not appropriate to simply use the theoretical section parameters based on elasticity assumption as a benchmark for comparison. According to your suggestions, I have supplemented the section compressive modulus and bending modulus under each case in Table 1 and Table 2, so that readers can make an intuitive judgment on the performance of each case.

Comment 3: Based on the above remark, if the sections do not have similar properties, how are they assessed on a fair basis?

Response: In actual engineering, the selection of the support parameters requires comprehensive consideration of the constraints of safety, economy, practicability and other factors. When the performance of the support meets the requirements, the designer will further consider the economics of the scheme, that is, which type of support can obtain better bearing performance at the same material cost. Research group 1 is designed based on this scenario, the fairness of each case is reflected in its economy.

On the other hand, in order to ensure the constructability and the quality of the shotcrete, the thickness of the shotcrete layer cannot be increased indefinitely. Consequently, the section height of the steel girder is also limited, will greatly limit the bearing performance of the support. Based on this scenario, research group 2 aimed to study the optimal cross-section form, under the premise that the section heights are equal, while limiting the other parameters that affect the bearing performance to be within a reasonable and comparable range.

Comment 4: 14 is not easy to visualize. Perhaps using thicker lines for all elements would solve this.

Response: Thank you for your suggestion. Figure 14 has been modified.

Comment 5: An overall check for standard English and clarity of expression is needed. At some points it is hard to understand what the authors mean to state, or can be misleading. For example, based on the manuscript, it seems that the authors meant to say “optimal” instead of “optional” in the introduction.

Response: Thank you for your criticism. The language of the article has been comprehensively polished and revised. As for the expression "optional", my original intention was to show that circular-shaped cross-sections are inferior to triangular-shaped cross-sections but better than other forms of cross-sections, so it is one of the options that can be considered. Since this article is a preliminary study, we try not to make too absolute judgments.

Reviewer 2 Report

The authors present an engineering problem that they analyze using FEM. Unfortunately, the model has a lot of simplifications, and secondly, the boundary conditions are not described clearly enough. The authors are aware of these simplifications, which they indicate in the discussion at the end.

The article is mainly aimed at Chinese people as 75% of the literature cited is in Chinese. Even with some justification, I believe this approach is questionable. 

The article is a bit boring, and the proposed solutions (triangular cross-section) may encounter many difficulties in implementation (also technological). Nevertheless, it is fairly well written.

I have marked a few comments in the file.

Author Response

Thank you very much for your attention and comments on our paper. We have revised the manuscript according to your kind advices. Our responses are as follows.

Comment 1: The article is mainly aimed at Chinese people as 75% of the literature cited is in Chinese. Even with some justification, I believe this approach is questionable. 

Response: Thanks for your criticism and suggestions. We have made a significant update to the literature review chapter. The research results of researchers from various countries have been summarized from the perspectives of theoretical analysis, numerical calculation, engineering applications, etc., which could supports the research of CFST support for traffic tunnels in this article.

Comment 2(at line 53): “What is the point of such an expression? Structures are always at home or abroad.”

Response: Thank you for your reminder. The original intention here is to express that the CFST is widely used in building structures in China and other countries, but this expression is indeed prone to unnecessary misunderstandings. The relevant expressions in the text have been revised.

Comment 3(at line 193): Why B=26cm? I don’t see any justification.

Response: When making the abstraction of the tunnel support according to the method in this article, the section width B of the abstracted member can be any value not greater than the distance between two adjacent steel tubes (of course, it cannot be too small to ‘cut’ the steel tube). This article mainly compares different cross-section forms of the steel tube, so the abstracted member should contain as little shotcrete as possible to highlight the role of steel tubes. In addition, considering that the cross-section of the steel tube used in the study has substantially equal dimensions in the cross-sectional height and width directions, the cross-sectional width B of the abstract member is taken to be equal to the cross-sectional height H.  We have revised the relevant expressions in the article.

Comment 4(at line 228): Sure, but debatable.

Response: Thanks for your kind advice. Strictly speaking, the bond-slip effect of steel tube and concrete at the interface should be considered, especially when using CFST as an axial compression member. However, when CFST is used as a bending member, studies have proved that considering the bond-slip has little effect on the overall mechanical response of the member. Moreover, adding discontinuities to the continuum-based numerical model will tremendously increase the computational cost. In summary, in the numerical model of this paper, the steel tube and concrete adopt ‘tie constraint’ at their interface. Relevant descriptions and citations have been added in the article.

Comment 5(at line 230): Not clear. Where you applied rigid support? Why would that simulate an anchor bolt connection? & What do you mean by ‘arch spring’?

Response: "Arch spring" refers to the foot of the arch member, which corresponds to the junction point between the crown and the bench when the tunnel is excavated by the bench method. Before the excavation of the bench, in order to temporarily ensure the stability of the support, foot anchors are usually applied at this point to provide horizontal and vertical restraint. Therefore, the foot of the abstracted member can be regarded as a fixed point. According to this assumption, rigid constraints are adopted to the nodes on the bottom face of the member. We have revised the relevant expressions in the article.

Comment 6(at line 235): In my opinion this is the most important part – unfortunately laconically described in the paper.

Response: Thank you for your criticism. A detailed description of the boundary conditions has been added to the article.

Comment 7(at line 238): Why uniformly distributed load? What kind of distribution?

Response: According to Chinese standards, the load on the deep-buried tunnel is generally simplified as a uniformly distributed load or a trapezoidal distributed load in vertical and horizontal directions, respectively. The members used in the calculation in this paper are abstracted from the crown of the traffic tunnel support. When there are no asymmetry factors such as bias pressure, the distribution of the load on the crown of the support is very close to the hydrostatic pressure state. Therefore, the uniformly distributed load is used to simulate the load of the tunnel support in the most general case.

Comment 8(at line 239): What do you mean?

Response: "Points to the arc center" intends to indicate that the direction of the uniformly distributed load coincides with the normal direction of the outer surface of the member. This statement in the article has been revised to make it clearer and more concise.

Comment 9(at line 640~644): It should be noticed also at the beginning of the paper.

Response: Thank you for your suggestion. The relevant description has been added in the introduction.

Round 2

Reviewer 1 Report

In the revised manuscript the authors have adequately addressed the reviewers' comments.